# Methods for the Calculation of the Lost Profit in Construction Contracts

**Radovan Majer \*, Helena Ellingerová and Jozef Gašparík**

Department of Building Technology, Faculty of Civil Engineering, Slovak University of Technology in Bratislava, Radlinského street 11, 810 05 Bratislava, Slovakia; helena.ellingerova@gmail.com (H.E.); jozef.gasparik@stuba.sk (J.G.)

\* Correspondence: radovanmajer555@gmail.com; Tel.: +421-903-421-890

**Abstract:** The aim of each investor is to procure the construction work in an efficient and economical way. This goal can be achieved by managing costs from the beginning of the investment process. It is necessary to determine the estimated price of the construction work in all phases of the investment process and not to underestimate the importance of this activity. It is almost a rule that the contractor or investor does not allow sufficient time for the contractor to prepare the construction for good quality, which may lead to insufficient preparation. The consequences of poor construction preparation vary from poorly built construction to litigation over the lost profits of the contractor—and this is the topic we discuss in this paper. The issue of asserting lost profits on the contract by the contractor is the subject of legal disputes between the contractor and the customer of construction work. In such cases, the question becomes the design of a methodology suitable for its calculation. The article deals with the presentation of the existing methods of loss of profit calculation, two of which are applied to the example of litigation from construction practice, with the definition of their results and differences.

**Keywords:** profit; calculating; calculation formulas; cost; construction; method

## 1. Introduction

Every year thousands of contractors go bankrupt partly because of the high level of uncertainty in the construction industry. Although many factors could be the cause of business failure, financial and budgetary factors are the most common causes [1]. More than 60% of contractor failures have financial causes. The lack of finance causes 77%–95% of contractor failures [2]. The absence of the linkage between financing and project scheduling affects the cash flow and creates nonexecutable schedules that lead contractors to a high rate of failure. Having financing problems not only affects cash flow, but also may influence the interactions among project participants. Conflicts among parties may increase [3], more claims may be filed [4] and contract failures may increase [2]. Thus, not only the integration of financing and scheduling, but also the minimization of financing cost of the project is of vital importance in managing construction projects successfully [5]. Profit represents for the company the amount of funds that will remain after the payment of all its payment obligations. Under the provision of Article 2 (3) (b) of Act of the National Council of the Slovak Republic No. 18/1995 Coll. on prices, when pricing a construction work, it is possible to calculate reasonable profit only. "Reasonable profit means profit based on development of the normal share of domestic goods profit in economically eligible costs, taking into account the quality of the goods, the usual risk of production or circulation and the development of market demand." What is the reasonable profit in construction contract? The starting point of this issue will be methods for calculating (quantification) reasonable profit on the contract, their analysis and review of established and new calculation formulas in terms of necessary background, advantages and disadvantages of their possible application. Based on this

specification and presentation of methods of profit calculating in Slovakia and foreign way, there is an elaborated proposal of determination of reasonable profit in contract in terms of construction experience. The whole research process is presented by a model case from a real situation and all of these methods can also be used abroad. However, the method of using valuation tools must be adapted according to the type of valuation tools used in that country.

## 2. Materials and Methods

Profit worked out by a company allows for development. New machinery can be acquired, contracts of bigger values can be won and financed. As a result of that, higher profit can appear in a company financial statement. A tendency to making profit higher and higher can be explained by safety of a company existence on the market. The significant part of profit usually is not spent, but left in a company as a reserve for crisis time. This is one of the most important reasons for making a profit—willingness to survive [6].

Appropriate profit should be calculated in the prices of construction works in order to ensure their competitiveness. Profit and risk calculations are processed transparently, based on data from the Statistical Office of the Slovak Republic and compliance with applicable legislation in the same way as indirect costs.

According to the Statistical Office of the Slovak Republic, the nationwide profit is 15.2% on average, for building development it is 18.2% and for civil engineering it is 13.1% [7].

After analysing this situation and having consulted several times, the guide-makers of indicative prices in the guideline profit calculated from price level 2018. We took into account the price range profit of 14%–16% in the construction works of buildings and civil engineering works [8].

At present, indicative prices of construction work cost are calculated with a profit by percentage surcharge of the processing cost, which is the sum of the costs of direct wages, operation of construction machinery, other direct costs, production and administrative overheads.

For the purposes of the company price calculation and calculation of the necessary% rate of profit, it is necessary to know the minimum amount of profit required. The share of reasonable profit largely affects the risk of construction work. This risk is influenced mainly by the degree of resolution of the project documentation, the quality and completeness of the tender documents, the nature of the construction, the conditions of implementation, delivery and payment conditions.

The necessary amount of the minimum necessary profit for the company can be determined by calculating the total need of funds necessary for the fulfilment of all managed funds (prescribed reserve fund and other voluntarily managed funds in the company). The expected credit volume resulting from necessary frontloading and advance spending for construction production shall be added. The necessary funds for own investments, extraordinary remuneration of employees, funds to cover unforeseen expenditure (elimination of risk in the process of realization of construction work) will be added. The amount thus obtained will be increased by the amount of depreciation and tax will be added, as profit as a source of income is subject to taxation in accordance with applicable legislation (for 2019: 19% self-employed, 21% legal entity).

Based on the knowledge of the competition, the amount of profit calculated in this way can be used to predict the necessary percentage rate for profit in the unit prices of construction works:

$$P = PROFIT/PC \times 100 (\%) \tag{1}$$

P—percentage of rate of profit
PC—processing costs

The presented method in profit calculations is not the final solution. For the needs of training it is necessary to put into practice a progressive tool of calculating objectification of profit.

## 2.1. Algorithms of Profit Calculation

The bases for calculation of profit may vary, depending on the calculation formula. It is important that the same base is used to calculate the percentage of rate of profit as well as the calculation of the amount of profit when calculating the unit price [9].

Algorithms for calculation of the amount of profit in unit prices of construction work are based on the used cost models.

These models also show the methodology of calculation of profit in unit prices of construction work, which is documented in the following diagram.

1. Under the traditional method of profit calculation or the Calculation Formula, the basis for calculating profits using a specified% rate of the processing costs, which shall be the sum of direct and indirect costs in the structure (Figure 1):

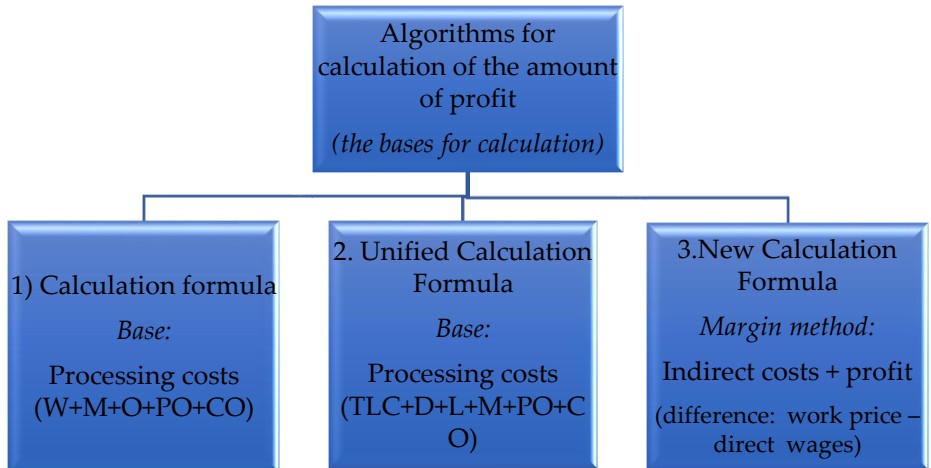

**Figure 1.** Basis for profit calculation.

Wages + machinery + other direct costs + production overhead + correct overhead

2. According to the Unified Calculation Formula, the processing costs also form the basis for the calculation of profit, but applying a different method of calculating indirect costs than that applied in the traditional calculation formula:

Total labour costs (direct wages + fringe benefits + levies) + machinery + production overhead + correct overhead)

3. According the New Calculation Formula, indirect costs and profits constitute a margin that is included in the unit prices at a specified% rate of direct wages

Margin = indirect costs (wage levies + production overhead + administrative overhead + profit). The method of calculating the margin is based on the difference between the price of labour and direct wages.

To better illustrate the described algorithms, we will give a few examples in the form of model calculations of profit in construction work "Contact thermal insulation system thck. 150 mm Weber-Terranova expanded polystyrene (EPS thickness 140 mm)" [10] according to the considered different bases for determining the profit.

In the methodology of calculating the profit resulting from Figure 2 and the traditional base of processing costs using the traditional calculation of indirect costs from the base Direct processing costs and Processing costs of production, the calculated profit in construction work is € 1.56, which is 15% of [C]+[L]+[M]+[O]+[OV1]+[OV2].



| | Price analysis | | | | |
|---|---|---|---|---|---|
| **Item** | 625252010 | Contact thermal insulation system hr. 150 mm Weber - Terranova (EPS), screw anchors | | | |
| | R000 | Decay - TOV 000 | | **MJ** | m2 |
| | | | | | |
| D | Direct material | | | | 28.41 |
| C | Labour costs + fringe benefits | | | | 4.19 |
| L | Levies | 35.2 % from wages | | | 1.47 |
| M | Machinery | | | | 0.00 |
| O | Other direct costs | | | | 0.00 |
| SUB | Subdelivery | | | | 0.00 |
| DPC | **Direct processing costs** | [C] + [L] + [M] + [O] | | | **5.66** |
| Overheads | Direct costs | [D] + [SUB] + [DPC] + [NC] | | | 34.07 |
| OV1 | production | 62.00 % from [C]+[L]+[M]+[O] | | | 3.51 |
| OV2 | administrative | 13.00 % from [C]+[L]+[M]+[O]+[OV1] | | | 1.19 |
| OV3 | | 0.00 % from [] | | | 0.00 |
| | **Indirect costs** | [OV1] + [OV2] + [OV3] | | | **4.71** |
| | **Total costs** | [D] + [SUB] + [DPC] + [OV1] + [OV2] + [OV3] + [NC] | | | **38.78** |
| P | Profit | 15.00 % from [C]+[L]+[M]+[O]+[OV1]+[OV2] | | | 1.56 |
| OV4 | | 0.00 % from [] | | | 0.00 |
| NC | Non-chargeable costs | | | | 0.00 |
| | **Total** | [D] + [SUB] + [DPC] until [NC] | | | **40.33** |
| | **Unit price** | | | | **40.33** |
| | Weight | | | | 0.0117 |
| | Standard hours | | | | 0.792 |

**Figure 2.** Calculated profit according to the Calculation Formula (CF) methodology.

In the methodology of calculating the profits arising from UCF (Figure 3), where the base for the calculation of overhead costs is made up of total labour costs, the calculated profit from the base of Processing costs in construction work is €1.46, which is 15% of [C]+[L]+[M]+[O]+[OV1]+[OV2].

| | Price analysis | | | | |
|---|---|---|---|---|---|
| **Item** | 625252010 | Contact thermal insulation system hr. 150 mm Weber - Terranova (EPS), screw anchors | | | |
| | 000 | TOV 000 | | **MJ** | m2 |
| | | | | | |
| D | Direct material | | | | 26.02 |
| C | Labour costs + fringe benefits | | | | 4.10 |
| L | Levies | 35.2 % from wages | | | 1.44 |
| M | Machinery | | | | 0.00 |
| O | Other direct costs | | | | 0.00 |
| SUB | Subdelivery | | | | 0.00 |
| DPC | **Direct processing costs** | [C] + [L] + [M] + [O] | | | **5.54** |
| Overheads | Direct costs | [D] + [SUB] + [DPC] + [NC] | | | 31.57 |
| OV1 | production | 62.00 % from [C]+[L]+[M]+[O] | | | 3.44 |
| OV2 | administrative | 13.00 % from [C]+[L]+[M]+[O]+[OV1] | | | 0.72 |
| OV3 | | 0.00 % from [] | | | 0.00 |
| | **Indirect costs** | [OV1] + [OV2] + [OV3] | | | **4.16** |
| | **Total costs** | [D] + [SUB] + [DPC] + [OV1] + [OV2] + [OV3] + [NC] | | | **35.73** |
| P | Profit | 15,00 % from [C]+[L]+[M]+[O]+[OV1]+[OV2] | | | 1.46 |
| OV4 | | 0,00 % from [] | | | 0.00 |
| NC | Non-chargeable costs | | | | 0.00 |
| | **Total** | [D] + [SUB] + [DPC] until [NC] | | | **37.18** |
| | **Unit price** | | | | **37.18** |
| | Weight | | | | 0,01184 |
| | Standard hours | | | | 0.792 |

**Figure 3.** Calculated profit according to the Unified Calculation Formula (UCF) methodology.

The aggregate margin rate (Figure 4) (levies + PO + CO + Profit) from the base of Direct Wages for the calculation of the indicative prices 2016/I is 148.00%.

| Unit price analysis | | | | | |
|---|---|---|---|---|---|
| price list: 011 | part of price list: A01 | | item code: 625254215 | TOV:000 | |
| legend: Contact thermal insulation system thck. 150 mm Weber - Terranova (EPS) | | | | | 29.07 EUR |
| description TOV: | | | | 0.00716T | 0.764Nh |
| Calculation formula ODIS (KV ODIS) | | | | | |
| 1. Material | | | | | 21.3 |
| 2. Work | Wages: 3.24 | | Rate in %: 248.00 | | 8.4 |
| 3. Machinery and equipme | Machinery: | | Other costs: | | |
| 4. Subdelivery | | | | | |
| 5. Constructions | Addition of separately valued items at group prices: | | | | |
| Price | | Price without VAT: | | | 29.7 |
| Costs breakdown | | | | | |
| Item code | costs description | | amount and m.j. | price for m.j. | total |
| 82113210 410011 | Drinking water for all consumers and producers of water + sewerage | | 0.00016 m3 | 1.89 | 0.0003 |
| 2831BS128 | Insulating board eps ISOVER 70 F - thck. 14 cm 1000x500mm | | 1.05000 m2 | 8.76 | 9.198 |
| 5538E0162 | Glass-textile grid 145g/m2 bal. 1650 m2 | | 1.1000 m2 | 1.17 | 1.287 |
| 5539L0139 | Screw plastic expansion tube with metal mandrel STR U 275-8/60x275 mm, pck. 1 | | 5.000 piece | 1.38 | 6,90000 |
| 585921000 266410 | WEBER-TERRANOVA Cement-based adhesive and reinforcing mortar | | 7.000 kg | 0.52 | 3,64000 |
| 7120-00-4 | Worker, HSV, tr.4 | | 0.35929 Nh | 3.97 | 1.42638 |
| 7121-10-5 | Mason, HSV, tr. 5 | | 0.39943 Nh | 4.51 | 1.80143 |
| 7129-01-3 | Construction worker, HSV, tr.3 | | 0.00481 Nh | 3.54 | 0.01703 |

**Figure 4.** Calculated profit according to New Calculation Formula (NCF) methodology.

For the example of construction work: Margin = €3.24 × €1.48 = €4.795.

As can be seen from the above, the profit is linked to the calculation of indirect costs in particular; therefore, any adjustment to the calculation of overhead costs will result in a change in the amount of profit. In these models, the basis for calculating profit "Processing costs" is traditionally considered without the inclusion of the item "Material" (mass). At present, the construction practice is considered to include material costs in the base. The profit and risk surcharge, expressed as a percentage rate of the selected base, should include, in addition to appropriate remuneration for the work of the company also general business risk, general construction and common warranty risks. English-speaking countries are characterized by the fact that all guide prices are published in price publications only at the level of direct costs and the decision on the amount of indirect costs and the amount of profit is left to the market exclusively by each company. This forces the company to choose, to the maximum possible extent, any arbitrary but reliable system of monitoring the level of its own costs and empirically use this knowledge for individual calculation of new contracts. In comparison, it can be seen that the new calculation formula is based on the simplified structure of the calculation formula used in the United Kingdom. Their calculation formula consists only of items that we refer to in our practice as direct costs (labour, materials and goods, machinery and equipment). The item "Work" contains basic salary rates, remuneration and other personal expenses. Both the Unified and New Calculation formulas are aimed at calculating the direct costs and indirect costs (overheads) as accurately as possible and profit must be agreed individually for each order, so they are adjusted according to the real competitive conditions in the given market segment of the region.

However, a loss of profit may also arise on the part of the investor in such cases when the contractor fails to complete the construction work in its entirety. Consequently, the investor cannot use the unfinished construction work for his financial benefit, while the expected financial efficiency of the construction work, which is usually calculated using simulation methods of the project investor before construction begins, differs significantly from reality, thereby generating lost profits for the investor [11]. In our paper, however, we focus on lost profits claimed by the contractor.

This can be a "change for the better" if it finally forces our construction companies to choose an arbitrary but reliable system of tracking their own cost levels and empirically use this knowledge to individually calculate the contract prices of new contracts, which may also be useful for process of public procurement [12].

Claiming a reasonable profit when the customer withdraws from the contractual terms.

In construction practice, there is no exception to a situation where the client (investor) unlawfully withdraws from the contract for work and the situation results in a lawsuit for lost profits applied from the contractor's tent. How is this category of profit defined in the legislative environment?

Act no. 513/1991 Coll. Section 3, § 379, refers to the claim for lost profits as follows: "Unless this Act provides otherwise, actual damages and lost profits shall be compensated. Damage exceeding that the liable party at the time of the contractual relationship arose as a possible consequence of the breach of its obligation was foreseen or could be foreseen in the light of facts that the liable party knew or ought to have known under normal care." [13].

The lost profits were defined according to the ruling of the Supreme Court, file no. 4 Cdo 319/2008, of 28 April 2010, as follows: "The loss of profits is detrimental in that the injured party did not, as a result of the loss event, have multiplied his assets, although this could have been expected with regard to the regular course of things. Loss of profits does not manifest itself as a reduction in the assets of the injured party (loss of assets as in actual damage) but in the loss of the expected benefit (return). It is not enough that the probability of the multiplication of property is sufficient, since it must be established that, in the course of a regular event (except for the unlawful conduct of an offender or a loss event), the injured party could reasonably expect an increase in his property that did not occur precisely as a result of the action of the offender" [14].

Accordingly, it follows from this definition that in the event of an unfounded or unjustified withdrawal from the contract of work, the contractor is entitled to this lost profits, which is calculated as the price by which the contractor would increase its sales if the contractor would complete the work in full.

The lost profits could be used within the estimated contingency reserve [15]. The contingency budget can be referred to as the amount of money within the cost baseline that is allocated for identified risks that are accepted and for which contingent or mitigating responses are developed [16]. Typically, to avoid deviations from the baseline finances [17], contingency budgets in complex projects are allocated either using qualitative and semi-quantitative techniques ([18,19]) or probabilistic [20,21]) and simulation-based methods [22].

*2.2. Methods of Calculation of Lost Profits in Our and Foreign Construction Practice*

Currently, the following methods are used to calculate lost profits:

2.2.1. The "Before and After" Method

This method compares, on the basis of accounting documents from the contractor, its profitability in anticipated sales and actual sales (Figure 5). Thanks to the given data it is possible to estimate what profits the contractor would have had in the absence of unlawful conduct on the part of the customer. This method takes into account all market trends and other impacts that may contribute to an inaccurate estimate [23]. Due to its simplicity, this method is one of the most widely used methods of determining the amount of lost profits abroad. However, it requires increased demands on managing internal records broken down to contract, not only for the selected accounting period. An example of this method can be seen in the following simple chart provided that the fictitious contractor company is working on only one contract for the client who withdraws from the work contract during construction.

The main disadvantage of this method is the increased demand for internal records of contracts. This means transparent dividing of contracts using groups of contracts and monitoring of contracts in the form of projects with the possibility of division into stages or phases of the project, including the possibility of financial evaluation. The advantage of this method is a very simple comparison of the planned profit with the actually achieved profit of the construction company.

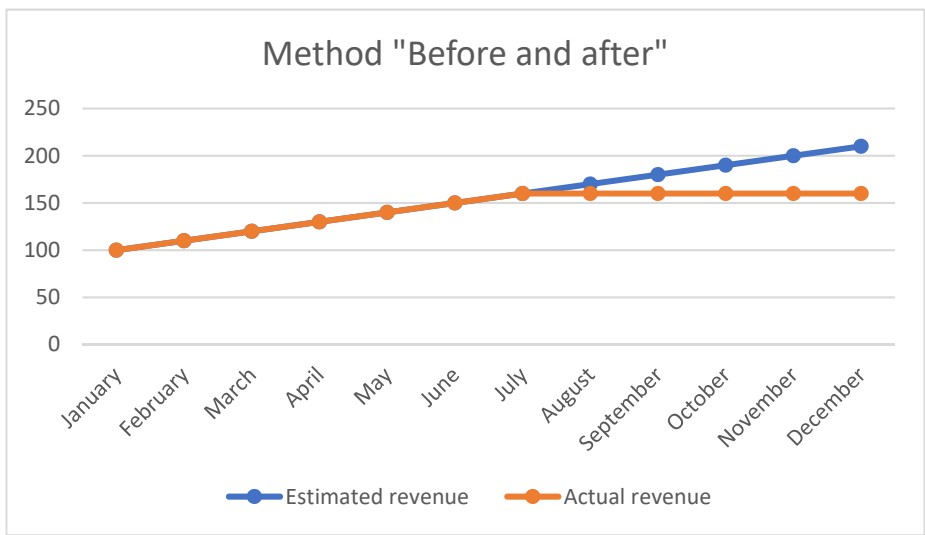

**Figure 5.** Graphical representation of the "Before and After" method.

### 2.2.2. Comparison Method (Yardstick)

This method is basically a comparative method since it uses a comparative analysis to calculate the lost profit. This method is used to compare the profitability and productiveness of other companies in the industry (in our case the construction industry). In this method, the loss of profit calculator determines the various specifics of the contract—e.g., contracting for technologically similar work and in a similar or same region, or based on overhead costs [24]. The calculation of lost profits by this method is based on the assumption that the injured company would carry out a comparable activity with other businesses if it were not injured; alternatively, the average of the profits of several companies having similar parameters to the damaged company may be used [23]. This procedure can be seen in the following Figure 6:

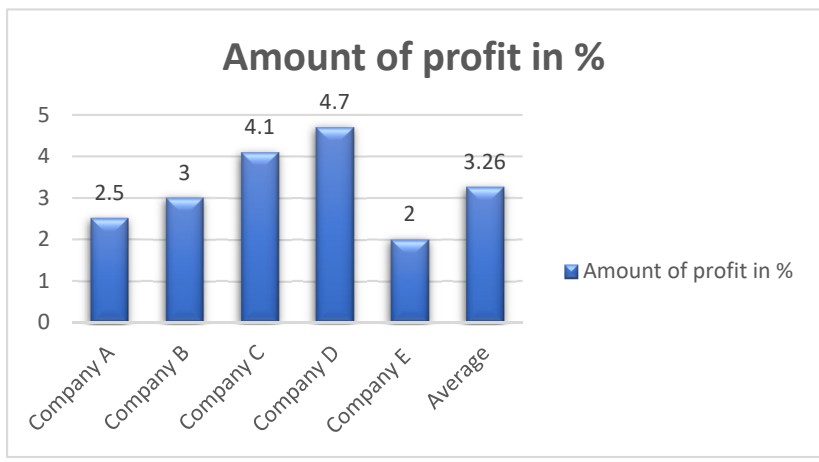

**Figure 6.** Graphical representation of inputs to the comparative method.

This comparative method is usually used in cases where we cannot reliably determine the profitability of the contractor to the extent that we use the Before and After method. The use of this method requires market research and knowledge of the management of companies in the construction sector.

The disadvantage of this method is the need of processing of comparative analysis, which can be laborious and the profitability data of other companies may not be accessible.

In the case of relevant inputs, there is the possibility of obtaining more objective outputs than the method before and after, since we compare a larger number of construction companies.

### 2.2.3. Method of Using Valuation Tools

Calculation allows you to track costs by the purpose of their spending and location (Figure 7). This method is already recognized as a traditional cost and profit calculation that provides information where costs arise and who is responsible for costs [25]. The most widespread method used in construction practice to quantify lost profits is the use of existing valuation bases and indicative price databases. For this purpose, it is possible to use valuation tools such as Cenkros 4 software, which uses a calculation structure of costs and a separate economic category-profit for the database of indicative prices of construction and assembly works. This calculation structure allows the calculation of the profit rate with a corresponding percentage of the total price of the construction contract.

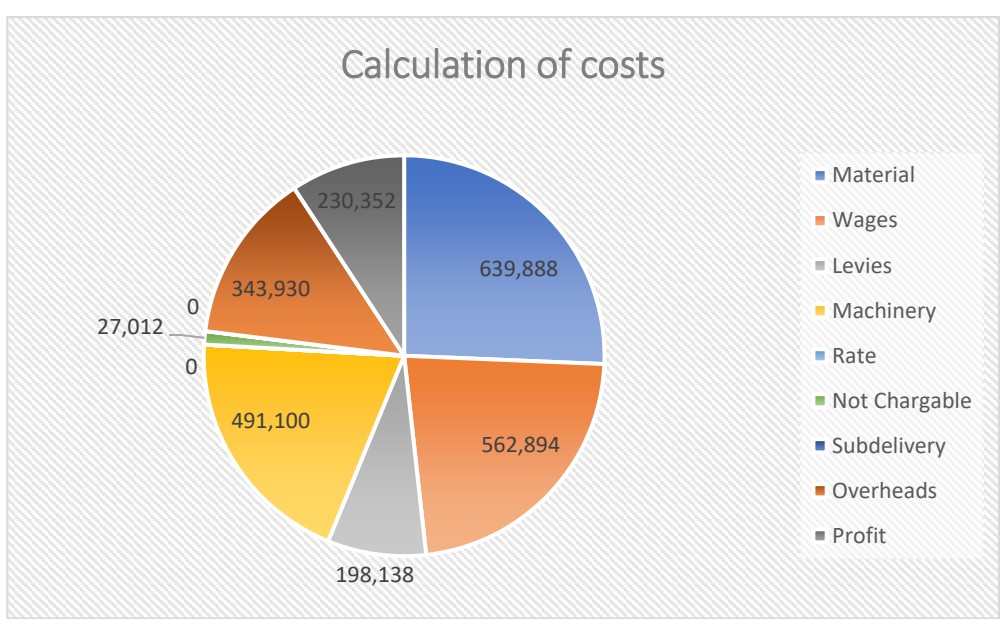

**Figure 7.** Displaying the calculation structure of the cost and profit on the contract (illustrative example) [26].

The disadvantage of this method is that we have to draw up a budget of completed and unfinished construction work, which can be laborious in large constructions.

The advantage of this method is the fairly accurate determination of profit according to established profit calculation practices in the country.

### 2.3. Model Example

When calculating lost profits on a model example of our construction practice, we applied the Comparison method "Yardstick" and the method of using valuation tools, specifically the software Cenkros 4. We could not use the method Before and after since we were not provided with accounting documents by the given company. We were unable to use the method of using the specific terms of service due to the fact that the contract for the work did not specify an agreed amount of profit on the contract or the method of calculating it (from what base and in what amount).

In the example given in Sections 2.3.1 and 2.3.2, the Contracting Parties agreed to carry out works worth EUR 50,122. The price of the work was agreed in the contract as a fixed price. The contractor carried out works worth EUR 17,842, which were invoiced and paid by the customer. Subsequently, the client became insolvent and the contract work was stopped. After two weeks, the Contractor withdrew from the Contract for Work and claimed lost profits on this contract.

### 2.3.1. Calculation of Lost Profits Using the Comparison Method

To use this method, we conducted a survey of sales and profits of construction companies with approximately similar parameters to the company in question. The survey results are shown in the following table (Table 1). Since the latest data was at the time of publication in 2017, we used these latest known statistics.

**Table 1.** Sales and profits after construction companies in 2017 [27].

| Company | Sales 2017 (Thousand Eur) | Profit After Tax 2017 (Thousand Eur) | Profit (%) |
|---|---|---|---|
| Strabag Pozemné a inžinierske staviteľstvo, s.r.o., Bratislava | 396,430 | 6,771 | 1.7080% |
| Doprastav, a.s., Bratislava | 290,979 | 29,355 | 10.0884% |
| Strabag s.r.o. Bratislava | 185,696 | 11,615 | 6.2548% |
| Eurovia SK, a.s. Košice | 167,522 | 112,317 | 67.0461% |
| Skanska SK, a.s. Bratislava | 111,425 | 595 | 0.5340% |
| Goldbeck, s.r.o. Bratislava | 103,126 | 6526 | 6.3282% |
| VUJE, a.s. Trnava | 90,038 | 5855 | 6.5028% |
| Váhostav-SK, a.s., Bratislava | 82,729 | 150 | 0.1813% |
| TSS Grade, a.s., Bratislava | 76,559 | 1470 | 1.9201% |
| Inžinierske stavby, a.s., Košice | 63,465 | 5827 | 9.1814% |
| DSC, a.s., Bratislava | 61,261 | 7336 | 11.9750% |
| Cesty Nitra, a.s., Nitra | 60,325 | 2981 | 4.9416% |
| Chemkostav, a.s., Michalovce | 59,306 | 737 | 1.2427% |
| HB Reavis Managment, s.r.o., Bratislava | 58,294 | −7395 | −12.6857% |
| Ingsteel, s.r.o., Bratislava | 57,083 | 675 | 1.1825% |
| AVERAGE PROFIT | 1,864,238 | 184,815 | 9.9137% |

The above data show that the average profit of comparable construction companies in 2017 was 9.91%.

Then the calculation of lost profit is based on the relation:

$$LP = Rp \times \mu/100 \tag{2}$$

where LP is lost profit, Rp is the residual price of work in progress, $\mu$ is the average profit of similar companies

After substituting the values into the formula, the estimated calculated amount of lost profit is:

$$LP = 32280 \times 9.9137/100 = 3200.14 \text{ EUR} \tag{3}$$

### 2.3.2. Calculation of Lost Profit Using the Valuation Tools Method (Cenkros 4)

For this method, we used the Cenkros 4 indicative price database with the choice of the price database II.Q/2017, due to the comparable price level at which the comparative method was applied (Section 2.3.1).

We applied the process in three steps:

1. At the given price level (in the program, in the database II.Q/2017) we prepared a budget of unfinished construction work on the construction.
2. From the budget, we found a calculated price of unfinished work of €32,280.
3. We used the function "display of calculation structure" in the program, shown in Figure 8. From this graph, we can determine the amount of contract profit calculated by the software Cenkros
4. Using the used calculation formula in the calculation structure of the cost and profit 10.25% of the residual price of work in progress, amounting to EUR 3309.

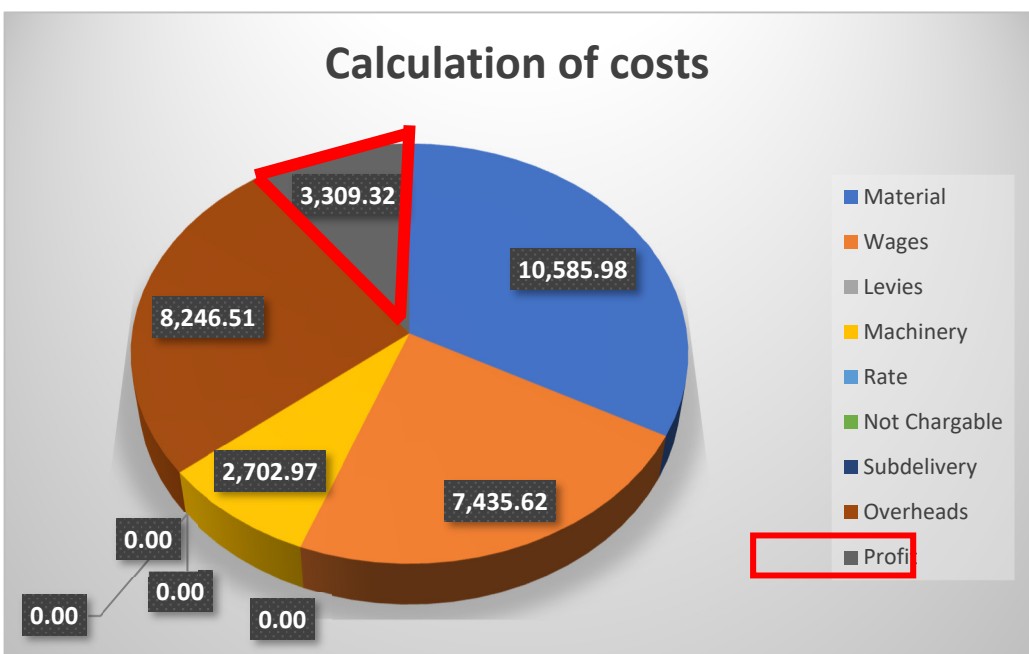

**Figure 8.** Displaying the calculation structure of the cost and amount of calculated profit on the contract [26].

When calculating lost profits by two different methods, we can observe the minimum difference between individual output values. While the loss of earnings calculated using the comparative Yardstick method is equivalent to EUR 3200.14, the loss of earnings calculated using the valuation tools method (Cenkros 4) is EUR 3309. Both methods have similar results and hence equivalent use in our construction practice. We recommend using both methods simultaneously for verification.

## 3. Results

When calculating lost profits by two different methods, we can observe the minimum difference between individual output values. While the loss of earnings calculated using the comparative Yardstick method is equivalent to EUR 3200.14, the loss of earnings calculated using the valuation tools method (Cenkros 4) is EUR 3309. Both methods have similar results and hence equivalent use in our construction practice. We recommend using both methods simultaneously for verification.

## 4. Discussion

At the present, the following methods can be used to calculate the lost profits:

- The Before and After Method using the contractor's accounting documents to estimate the approximate percentage of profits. Due to the simplicity of calculation, this method is one of the most widely used abroad. However, it requires increased demands on the evidence of the economy divided into a specific construction contract.
- The Comparison Method (Yardstick) uses a comparative analysis to calculate the loss of profit, the principle of which is in comparison with the profitability of comparable companies.
- Method of using valuation tools calculates the lost profits by using valuation tools. Software Cenkros 4, ODIS Žilina or Calkulus.

In the considered model example, the calculation of the lost profits is applied by using two selected methods. The output of the application proves that the calculation of the lost profits by the Comparison Method (Yardstick) and the method of using valuation tools achieves very similar results. Despite this fact, we recommend always using at least two methods in the calculation to verify the results. These

methods may also be useful in expert examination and evidence, in an area where contractors are increasingly required to determine the amount of the lost profit on the contract.

## 5. Conclusions

Profit is a specific economic category among the partial components of the price. By definition, profit is a surplus-value derived from an economic activity in proportion to the contributed capital. It is not only a "reward" for business, but also a condition for further development of science and research, modernization, improvement, above-standard social care for employees, etc. Profits are generated by sales that exceed the costs. In cases where costs exceed sales, there is a loss. The results of this study are beneficial for further application of calculations in expert activity. The construction experts are most often asked to determine lost profits on the construction contract and in the process of litigation between the client and the contractor. Thus, construction experts can reliably use these methods in determining of lost profits. It is clear from the paper that they must use at least two of the methods described, precisely to verify the outcome.

Another benefit of this study is the application of its results, which can be used by contractors and customers in case they decide to apply amicable settlement when they withdraw from the contract. Both parties can therefore follow this procedure and thus at least approximately calculate the lost profits of the contractor on a particular construction contract. In the context of the current situation, we are unlikely to avoid the global economic crisis, which will also affect construction. The methodology for determining a reasonable loss of profit in construction contracts will also find application more than ever.

Profit fulfils important tasks:

- it is a criterion for deciding on the economy of a company;
- it is a tool for accumulating and creating funds for the development of the company;
- it is the basis of redistribution between the business and the state;
- it motivates every entrepreneur.

The following criteria must be taken into account when calculating profit:

Profit level is the result of supply and demand. The amount of profit depends on the strategy of the company, the market environment and the competitiveness of the company in the market. Reasonable profit is the minimum profit that the company will ensure the development of its production program, its own existence and good prosperity. The restrictions of the current study concern, in particular, small construction contracts and smaller-scale reconstruction works, the inputs of which were available to the authors of the study. The restrictions also apply when the parties do not conclude a written work contract between themselves. These works tend to be of a lower price, so for this reason it is assumed that upon withdrawal of the work contract, the parties will agree without the need for third party or court intervention. However, practice is proof that this is not a correct assumption and disputes over the payment of a relatively "lower value of lost profits" have been resolved by lawsuits for several years. Therefore, the authors of the study began to address the issue just on smaller construction contracts, with relatively low price values. However, they could be ultimately liquidated for the company.

In the near future, we would like to extend our study not only to the loss of profit calculation methods applicable to large construction contracts, but also to focus on another group of problematic construction costs, such as overhead costs, i.e., costs that are indirectly ensuring the operation of construction production.

**Author Contributions:** H.E. wrote the first draft of the article, describing the issue in detail. J.G. carried out a survey of the issue abroad and incorporated a copy into this article. R.M. performed a statistical analysis of the data and then interpreted it using a calculation. All authors have read and agreed to the published version of the manuscript.

**Funding:** This research was funded by project VEGA, grant number 1/0511/19.

**Acknowledgments:** We would like to thank our university for providing the opportunities and resources that have helped us to create this article.

**Conflicts of Interest:** The authors declare no conflict of interest.

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
