# Peer review of "Methods for the Calculation of the Lost Profit in Construction Contracts"

_buildings, doi:10.3390/buildings10040074_

Round 1

Reviewer 1 Report

  1. The study concerns the profit topic in the economic activity.
  2. First a profit fulfils tasks list was identified and characterized as a support to defined a list of criteria that must be taken into account when calculating profit. These remarks are important to the community.
  3. The methodology and the bibliographic support are adequate.

Reviewer 2 Report

The calculation of profit in construction is definitely an important topic. But the research question and methods are not clear for me. The paper lists different ways of propfit calculation, but the advantages and disadvantages of the different methods are not well described. From my point of view this has to be improved. 

The success of failure of managing construction has not only financial issues. What´s about specifications, documentations, knowledge management, standards, data collections and process management?

Reviewer 3 Report

The topic of this study is interesting. The study looks at main existing methods used for calculation of the contractor's loss of profit when customers' withdraw from their established contracts. The title however does not reflect this and can be confusing. The title suggests that the main focus of the study would be on ways that contractors could adopt for estimating 'reasonable' profit in their bids for projects. So as a suggestion, the title can be amended to be 'compensation for profit-loss' as opposed to 'reasonable profit'.

The study starts by providing a summary on general methods used for calculating profit in construction contracts, which is useful. It then uses illustrative examples to show how calculations can be conducted and the difference in results between the methods. This section of the paper requires major revision and more clarity on the contents of Figures 2 and 3. For example, how have the calculated profits been calculated. This section also lacks a conclusion. It can also be improved by explaining how it links to the following section (i.e. calculations of profit-losses). Further, the study keeps referring to our country and foreign countries, without explicitly indicating this at introduction and in the title of the study itself - That the focus is on Slovakia - and why? 

The study then summarises three wide-spread methods used for calculating contractor's profit-losses if a client chooses to withdraw or terminate a contract and not complete a project. The study then uses '1 simple' model example of a project of a very small project value (50,000 Euros) to compare between the results of two of these methods, and concludes that there is little difference in the output values. 

The paper is useful. However it is more like a report. It is unclear how the study makes new contribution to existing knowledge. Additionally the methodology used for the study is very limited. The paper also requires significant editing and major revision to English language.

Round 2

Reviewer 2 Report

Thank you for adding disadvantages to the different approaches. 

I am still wondering that no standardisation exists for the methodology of cost calculation. If there are, then I am missing their references. 

Reviewer 3 Report

The author(s) have taken some useful steps to improve the paper. For example, adding a discussion about the disadvantages of each valuation methods has helped to improved the criticality of the arguments presented in the paper. There are however a number of recommendations that still need to be addressed:

1- The conclusion section needs to emphasise the main contributions of the study. How has this study contributed to existing body of knowledge? The conclusion section should answer this question more explicitly.

2- The paper needs to include a section towards the end in which it reflects on the limitations of the current study, and provides suggestions for future studies and directions.

3- The title modification is useful. It is suggested as follows: Methods for the calculation of lost profit in construction contracts

4- A final check and thorough revision and editing of any English grammar or typo errors needs to be conducted.
